# Honeybees adapt to a range of comb cell sizes by merging, tilting, and layering their construction

**Golnar Gharooni-Fard[1], Chethan Kavaraganahalli Prasanna[1], Orit Peleg** [1,2,3]*, **Francisco López Jiménez** [4]*

**1** Department of Computer Science, University of Colorado Boulder, Colorado, United States of America,
**2** BioFrontiers Institute and Departments of Physics, Applied Math, and Ecology and Evolutionary Biology,
University of Colorado Boulder, Colorado, United States of America, **3** Santa Fe Institute, Santa Fe, New
Mexico, United States of America, **4** Ann and H.J. Smead Department of Aerospace Engineering
Sciences, University of Colorado, Boulder, Colorado, United States of America

\* Orit.Peleg@colorado.edu (OP); Francisco.LopezJimenez@colorado.edu (FLJ)

London, UNITED KINGDOM OF GREAT BRITAIN
AND NORTHERN IRELAND

**Peer Review History:** PLOS recognizes the
benefits of transparency in the peer review
process; therefore, we enable the publication of
all of the content of peer review and author
responses alongside final, published articles.
The editorial history of this article is available
here: https://doi.org/10.1371/journal.pbio.
3003253

**Data availability statement:** The data used in
this study can be accessed on

## Abstract

Honeybees are renowned for their skills in building intricate and adaptive combs that display notable variation in cell size. However, the extent of their adaptability in constructing honeycombs with varied cell sizes has not been thoroughly investigated. We use 3D-printing and X-ray microscopy to quantify honeybees' capacity in adjusting the comb to different initial conditions. Our findings suggest three distinct comb construction modes in response to foundations with varying sizes of 3D-printed cells. For smaller foundations, bees occasionally merge adjacent cells to compensate for the reduced space. However, for larger cell sizes, the hive uses adaptive strategies such as tilting for foundations with cells up to twice the reference size and layering for cells that are three times larger than the reference cell. Our findings shed light on honeybees adaptive comb construction abilities, significant for the biology of self-organized collective behavior, as well as for bio-inspired engineered systems.

## Introduction

The ability of animals to construct nests that are well-adapted to their environment is crucial for their survival and reproduction. Across diverse taxa, from birds meticulously weaving nests [1] to termites engineering vast mounds [2,3], we observe remarkable flexibility in building strategies in response to both internal needs and external environmental factors [4, 5]. This adaptive construction often involves optimizing resource use (*e.g.*, caddisfly larvae selecting locally abundant materials for their cases [6]), improving functional performance against environmental challenges (*e.g.*, termite mound thermoregulation using diurnal temperature oscillations [7]), and employing modular or iterative building processes and repairs (*e.g.*, weaver bird communal nest structures [8,9], or ants that alter their nest structure to slow the spread of diseases during colony expansions [10]).

Honeybee nests are the emergent outcome of the collaborative efforts of thousands of individual bees. These structures play a vital role in the survival and functioning of a bee colony

https://datadryad.org/dataset/doi:
10.5061/dryad.z8w9ghxmw. The code used to
analyze the X-ray data in this study can be
found on https://zenodo.org/records/15610300.

**Funding:** This work is supported by a grant
from the CU Boulder Research and Innovation
Office Seed Grant Program to O.P. and F.L.J., by
NSF grant 2210628, both to O.P. and F.L.J., and
by BioFrontiers Institute internal funds to O.P.
The funders played no role in study design, data
collection or analysis, decision to publish, or
preparation of the manuscript.

**Competing interests:** The authors have
declared that no competing interests exist.

by providing storage space for nectar and pollen, a nursery for brood development, and a structurally stable environment for the various interactions among the nestmates [11,12]. The hexagonal shape of honeycomb cells, built with a near-optimal minimization of the wax-to-storage space ratio [13–15], has fascinated scientists for millennia [16–18]. This optimization is necessary because wax production incurs a high energy cost [18]. In particular, to secrete just 1 *lb* (454 *g*) of wax, honeybees need to consume about 8 *lbs* (3.8 *kg*) of honey [19].

Feral bees have been shown to construct their nests on a range of surfaces such as tree branches and pre-existing cavities, which do not necessarily allow for regular hexagonal lattices [20]. Furthermore, comb building is a distributed process with no global blueprint or a leader for bees to follow. Instead, workers often initiate constructing the comb simultaneously from multiple locations within the hive [21]. To achieve a coherent structure, separate pieces of comb need to be merged and workers must reconcile the differing geometries of cell sizes at the boundaries (notably in terms of size and alignment), while conforming to the available space and resources [22]. The resulting structure inevitably contains non-regular hexagons and topological defects with various cell sizes, to adapt to environmental constraints [21,23–25]. Adjusting the shape, size, and arrangement of hexagons during honeycomb construction may serve multiple purposes, often dictated by the environment or colony's requirements. Previous studies across various honeybee species have shown the precise design of their nest, optimized to maximize storage capacity and brood space [19,26,27]. These adaptations occur as the needs of the colony fluctuate, requiring more or less space for the rearing of the brood or the storage of honey over time [21]. Social wasps have also been shown to use architectural solutions to build their nests, similar to honeybees [28,29]. For instance, the combs of paper wasps exhibit remarkable adaptability, with structural variations influenced by factors such as nest maturation, overall size, and cell arrangement [28,30,31].

More than four decades ago, H. R. Hepburn [19] studied the comb construction by African Honeybees *Apis mellifera adansonii*, providing them with sheets of beeswax foundations having a range of cell sizes from 170 to 1022 *cells*/$dm^2$ (between 1 to 6 times the average size of the natural honeycomb cell). Using two-dimensional (2D) images of the built comb in the experimental frames, the study reported various construction solutions and measured the working tolerances of the comb with variable cell sizes. Yet, honeycomb represents a three-dimensional (3D) structure, and the details and intricacies of it cannot be fully discerned and quantified using 2D images. While the specific behavioral rules of individual bees during comb building remain unclear, revealing the 3D structure of the comb built under various initial conditions will be a crucial step in linking the local behavior to the global building patterns. For instance, the 3D progression of the nest building process, shown in a recent study by Marting et al [32], reveals that bees prioritize building a well-connected nest of parallel combs that remains structurally and functionally robust, even when faced with experimental disruptions. Such findings show the importance of accounting for 3D comb structure and provide valuable insights into the adaptive and resilient nature of the honeybee superorganism.

In this work, we study the adaptability of the comb construction by European honeybees *Apis mellifera* L, under different initial conditions, using 3D visualization. We leverage 3D-printing to design repeatable experiments with precisely controlled and carefully quantified cell areas as foundations. By exploring the responses of honeybees to various initial conditions (using foundations with different cell sizes), our objective is to exhibit the range of construction modes bees use to build their nest, as well as the complexity of these structures and their sensitivity to initial conditions. Our findings corroborate and expand upon prior research by delineating the distinct construction modes employed by bees in response to specific cell sizes. To highlight the details of each construction mode, we use X-ray microscopy (XRM)

to quantify the properties and visualize these intricate structures. In a recent study, Franklin et al. [33] have also used XRM technology to describe a step-by-step process by which honeycomb is constructed in natural settings, which sheds light on the structure of a natural honeycomb cell as it is being built. Their findings contribute to understanding the nest construction mechanism by showing how individual cells grow directly from the hexagonal pattern on the comb's spine, enhancing structural stability through staggered lattice patterns and strategic material deposition. Our work examines how foundations with different cell properties and arrangement give rise to distinct patterns and how these patterns can be interpreted in relation to the initial conditions on which they are built.

To distinguish various foundations, we define the cell size variable ($S$) as a multiple of the inner area of the average honeybee worker cell, that is derived from the length of a regular hexagon, which we measured to be 5.4 mm on a free comb built without a foundation. Fig 1A shows a schematic of the shapes of the cells built on the plastic base with $S = 1$, equal to the size of an average worker cell. Fig 1B shows two snapshots of a 3D-printed frame with $S = 1$ at the start and end of the comb building process. Fig 1C shows the 3D-visualization of the structure of the comb using X-ray data from a section of the frame shown in panel B. The details of how we process the raw X-ray data to generate 3D-visualizations of the comb structure are described in the Methods Section. Using $S = 1$ as our control, we supply the hives with custom 3D-printed frames with regular hexagonal patterns of various cell sizes, ranging from half to four times the size of a reference cell (*i.e.*, $0.5 \leq S \leq 4$). Our goal in changing the 3D-printed cell sizes is to examine the bees' capacity to adjust the comb structure to the given foundation. Fig 1D provides an overview of our results, highlighting different construction modes observed in our 3D-printed frames. It includes photographs of the obtained structures followed by schematics of adaptive strategies used for honeycomb construction with varying cell sizes.

## Results

We begin this section by discussing the different construction modes identified in our experimental data, followed by a more detailed description and visualization of the overall structure of the comb in each mode. We start our exploration by reducing the area of the 3D-printed cells on the foundations to half of the average worker cell size (*i.e., S* = 0.5). Our findings suggest that when faced with such small foundations, bees do not follow the patterns or use the printed cells in any other form for comb building. The leftmost column in Fig 1D shows that bees fill most of the small printed cells with wax and build new hexagons on a mostly flat foundation. On the other end, increasing the size of the printed cells to four times greater than the size of the average worker cell ($S = 4$) does not produce any consistent patterns (see the rightmost column in Fig 1D. However, within the range of $0.75 \leq S \leq 3$, we identify three different building modes—merging, tilting, and layering—relative to the size of the printed foundation. The intricate properties of each construction mode, along with 3D-visualization using XRM, are presented in the following sections.

### Merging

When faced with foundations where cell sizes are 0.75 times smaller than the size of an average worker cell ($S = 0.75$), bees adapt the comb structure and use the foundation with an unexpectedly intricate approach. Fig 2A–2C illustrates the progression of comb construction on one such experimental frame over a 20-day period. There is an apparent pattern at work from the early stages of comb construction, indicating the swarms' recognition of and adaptation to this new cell size pattern. As shown in the insets of Fig 2A–2B, even though

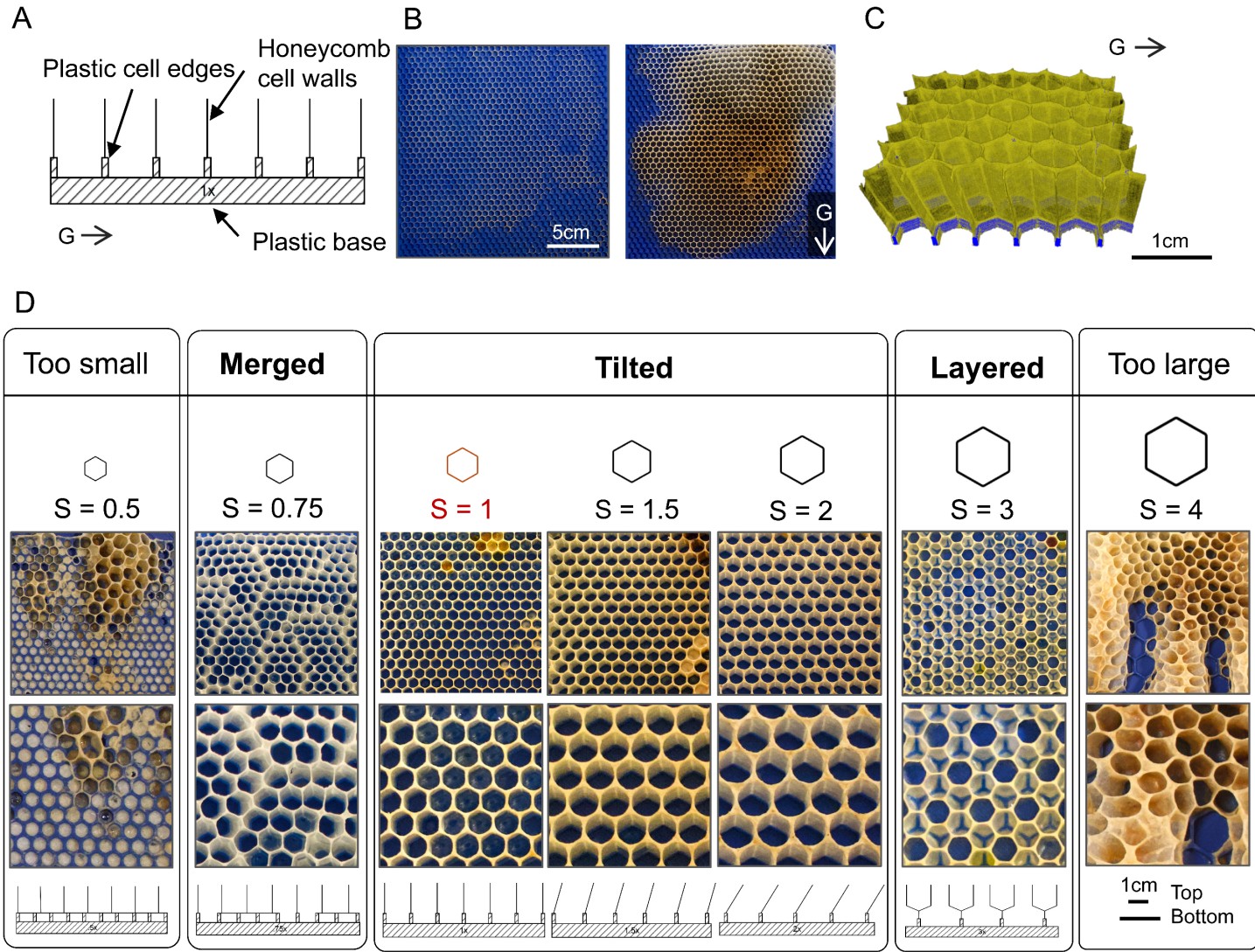

**Fig 1. Overview of results for different values of $S$.** A) Schematic of the honeycomb cells built on a control frame with $S = 1$. B) Two snapshots of the initial frame with the plastic, 3D-printed foundation along with the same frame after 20 days. The printed cell size on this frame is $S = 1$. C) XRM-based 3D-visualization of a section of the frame with $S = 1$, segmented to highlight the comb in dark yellow and the plastic in blue. The direction of gravity is indicated by the arrow labeled $G$. D) Overview of the different modes of honeycomb construction on 3D-printed frames with varying cell sizes. The top row shows the building mode in relation to the cell sizes on each foundation, displayed beneath them. More 2D images of all the built frames used in this study, organized by the 3D-printed cell size, can be found in https://datadryad.org/dataset/doi:10.5061/dryad.z8w9ghxmw. The subsequent row presents images of the built honeycomb on the foundations, accompanied by a zoomed-in view beneath. The bottom row illustrates schematics depicting adaptive strategies for honeycomb construction with differing cell sizes.

most of the drawn hexagons start from the plastic edges on the printed foundation, some cell walls are built with an angle that eventually merges with the walls of adjacent cells built with a reverse tilt. This pattern repeats until the entire frame is covered with hexagonal cells. The term "merging" describes the occasional combination of several small cells to create the necessary space to construct natural worker-size hexagons.

Fig 2C shows the frame after comb construction is complete, but prior to filling of cells with honey or brood, at which point X-ray imaging is conducted. We find that the merging behavior during honeycomb construction is not specific to our experimental frames with

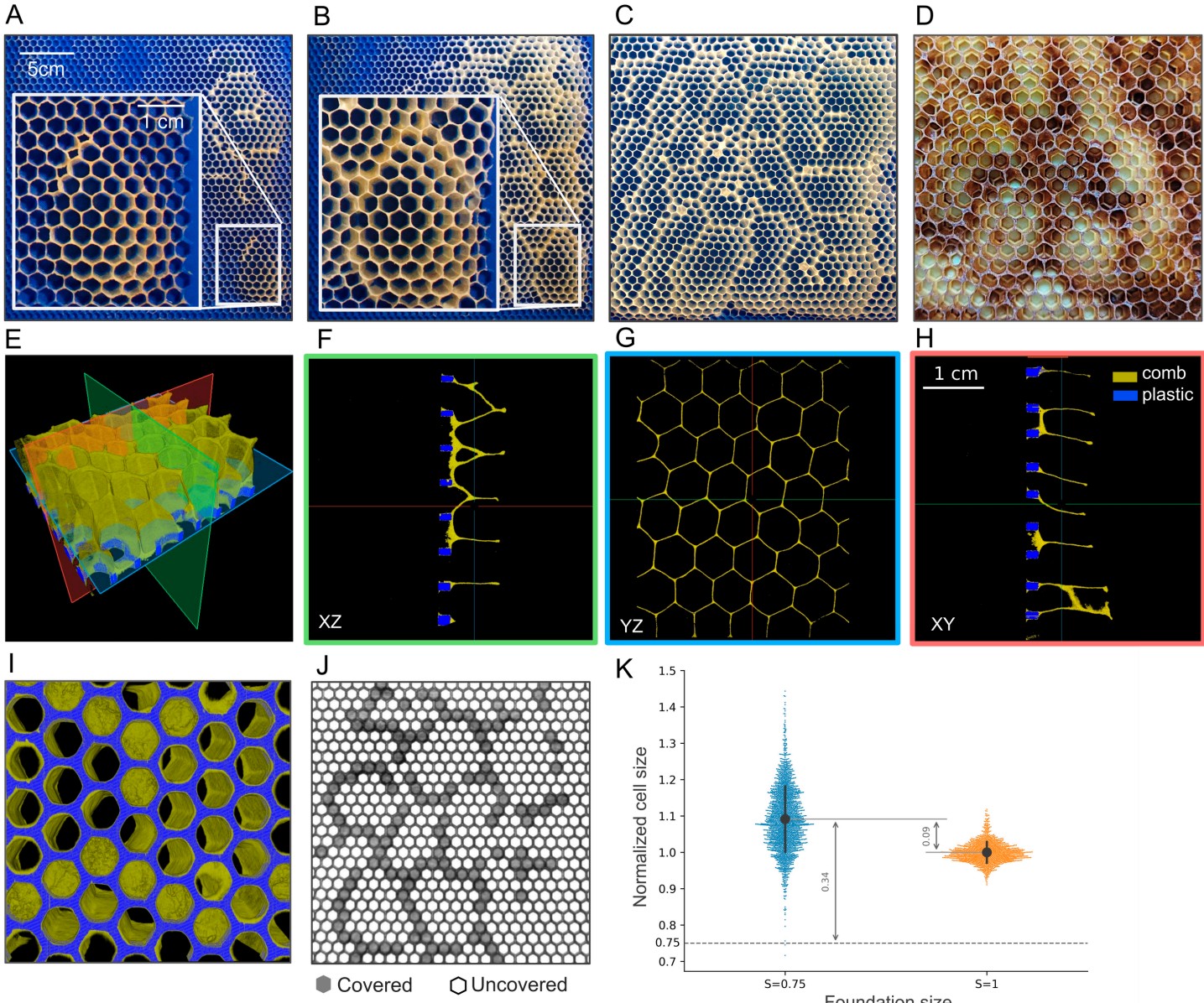

**Fig 2. Results of the comb built on samples with S = 0.75 foundation.** A) The 3D-printed frame of cells with $S = 0.75$ thinly coated with beeswax after 3 days of being in the hive. The inset shows the zoomed-in section of the constructed comb highlighting cells with tilted walls. B) The same frame after a week. The inset shows the same, but zoomed-in section shown in panel A, demonstrating how the adjacent cells with tilted walls will be merged once they are fully built, effectively concealing the smaller cell below on the foundation. C) The same frame after 20 days. D) A section of the honeycomb built on one of the non-experimental frames shows merging cells. E) 3D visualization of a section of the comb intersected by three planes. F–H) The XZ, YZ, and XY planes shown in green, blue and red. The comb is shown in dark yellow while the plastic is in blue. I) The bottom view of the 3D reconstructed volume showing the cells that are covered and uncovered cells. J) 2D visualization of a larger section of the same sample with covered cells highlighted in gray. K) Comparison of the normalized cell size distributions of cells built on foundations with $S = 0.75$ in blue and cells built on the reference frames with $S = 1$ in orange. The gray circles within each distribution indicate the mean for the distributions and the gray errorbars depict the standard deviations. The deviation in the mean cell size of cells built on $S = 0.75$ foundations is shown relative to both the reference cell size and the printed $S = 0.75$ cell size. The data underlying this figure can be found in https://datadryad.org/dataset/doi:10.5061/dryad.z8w9ghxmw.

smaller printed foundations. A similar pattern can sometimes be observed on the comb built on some of the commercially available frames with worker-size foundations (equivalent to our 3D-printed frames with $S = 1$), see S1 Fig. Fig 2D shows an image of the comb built on one

of the commercial frames inside our hives using the merging mode. This adjustment may be made to meet the colony's specific needs for constructing larger cells, such as those used for raising drones. The series of images in Fig 2E–2H demonstrate the X-ray microscopy results. In particular, Fig 2E and S1 movie illustrates a 3D-reconstruction of a 5 cm × 5 cm section of the frame in Fig 2C, highlighting plastic in blue and comb in dark yellow. To better visualize the merging process, the cross-sections of the X-ray are presented in Fig 2F–2H, using planes containing two of the Cartesian axes, shown in green (XZ), blue (YZ), and red (XY) in Fig 2E. For example, the XZ slice in Fig 2F shows how the comb material is used to join adjacent plastic edges beneath the cells.

Remarkably, while some of the printed cells are filled with wax, the top view of the comb structure, shown in Fig 2G, still features a regular hexagonal grid with cell sizes that are comparable to the average worker cell size. Fig 2K depicts the distribution of normalized cell size for cells built on frames with $S = 0.75$ in blue, and the same for reference cells ($S = 1$) in orange. The normalized cell size is computed by dividing the built cell areas by the mean reference cell area. For better interpretability, we will use normalized cell sizes in future discussions, unless otherwise stated. As depicted in Fig 2K, the mean normalized cell size built on foundations with $S = 0.75$ is 0.09 units larger than the mean reference cell size, but 0.34 units larger than the area of the corresponding printed cells $S = 0.75$. This suggests that on foundations with $S = 0.75$, bees are building honeycomb similar to the typical cell size that is closer to the reference cells than to those on the printed foundation. Moreover, apart from being slightly larger than the reference cell size ($\mu_{S=0.75} = 1.09$, $\mu_{S=1} = 1$, $t = 44.0385$, $P < 10^{-5}$ for a one-tailed two-sample Welch's unequal variances t-test), the variation in cell size for these cells is considerably wider ($\sigma_{S=0.75} = 0.09$, $\sigma_{S=1} = 0.03$, $W = 971.173$, $P < 10^{-5}$ for the Levene test to compare variances), suggesting less consistency in the cell sizes of the comb built on the smaller foundations. The higher standard deviation could be attributed to the discrepancy between the printed and desired cell sizes and the inherent randomness of the merging process. See Table A in S1 text for more details on the number of samples in each category, the individual sample size and statistics.

Additionally, in the bottom view of the sample in Fig 2I, it can be observed that some of the printed cells are entirely covered with comb. This coverage potentially aims to provide additional space for the initiation of the hexagonal lattice, with $S \approx 1$, which, as noted earlier, is what bees construct on the $S = 0.75$ foundations by merging cells. The distribution of covered printed cells on a larger section of the frame can be viewed in Fig 2J, with covered cells shown in gray. If we hypothesize that bees compensate for the 25% reduction in space per cell (relative to their average worker cell size) by combining cells in our 3D-printed foundations with $S = 0.75$, we would expect them to cover 25% ($h_0 = 0.25$) of the cells in the printed foundation to build a hexagonal lattice with $S \approx 1$. Interestingly, all individual experimental samples, as well as the pooled data, fail to reject our null hypothesis ($h_0 = 0.25$) under a two-tailed one-sample z-test for proportions at a significance level $\alpha = 0.05$ (when pooled together: $h_{pooled} = 0.24$, $z = -0.744$, $P = 0.457$). For details about the individual replicates, see Table B in in S1 text. This suggests that bees employ the merging strategy to effectively transform a slightly smaller hexagonal grid ($S = 0.75$) into their commonly used size ($S = 1$), and the behavior has been consistent across our experiments.

## Tilting

The next mode of construction corresponds to the case with the printed foundation of cells with $1 \leq S \leq 2$, where bees use the provided patterns but increase the tilt angle of the built cells as the cell size increases. Fig 3A presents a series of 2D images of the comb built on

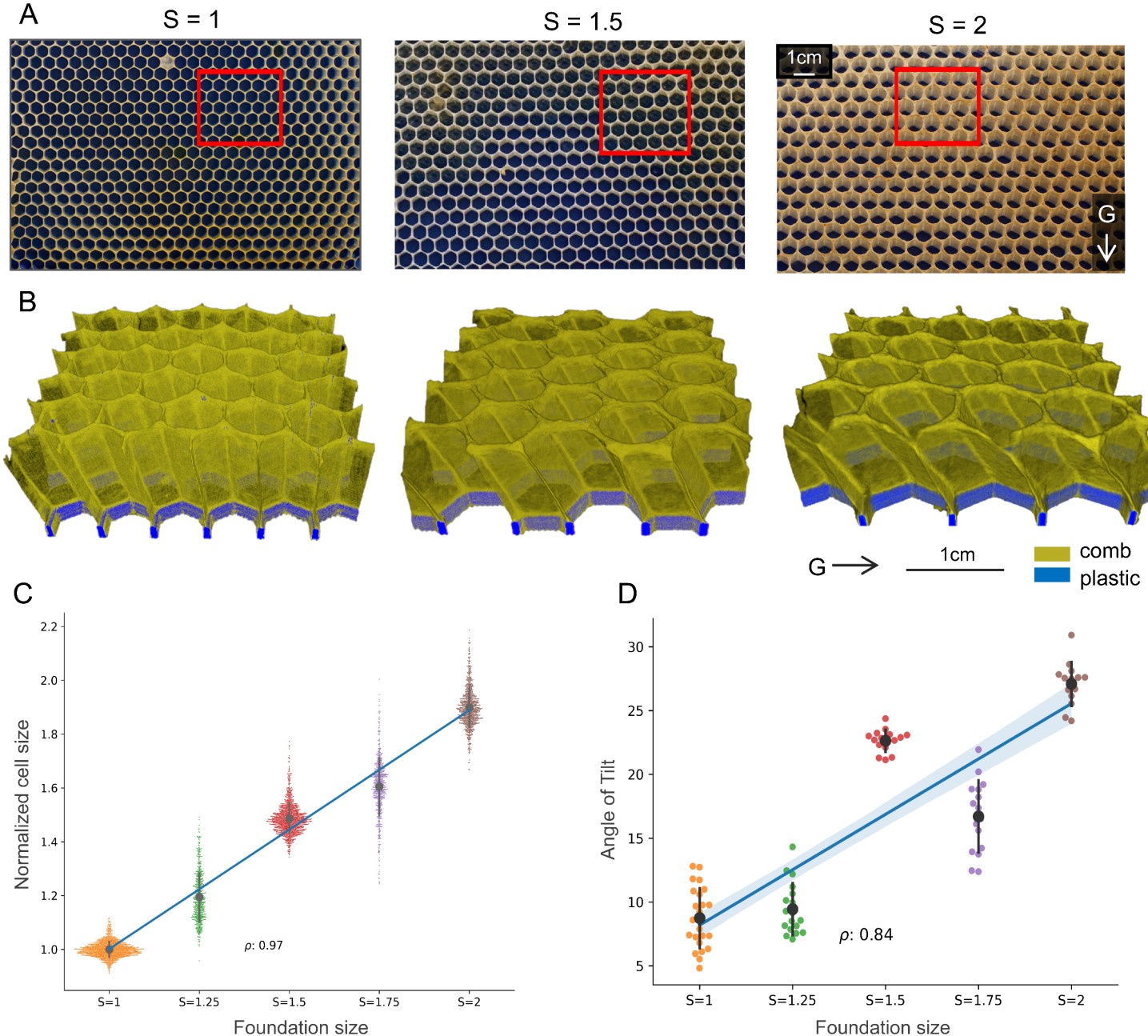

**Fig 3. Results of the comb built on samples with** $1 \leq S \leq 2$ **foundation.** A) The 2D images of the frame with $S = 1, 1.5$, and 2. B) The 3D reconstruction of the sections of the comb, highlighted with a red box in panel A, using X-ray data, segmented to show comb in dark yellow and the plastic in blue. C) Distribution of the normalized cell sizes (C) and angles of tilt (D) of honeycomb cells (vertical axis) built on the frames with foundation size $1 \leq S \leq 2$ (horizontal axis), colored by foundation size. Wider spread in the swarm of points indicates a higher number of data points at that normalized cell size. The gray circle within a swarm indicates the mean for that distribution with the gray error bars depicting the standard deviation. The blue lines in (C) and (D) indicate linear least-squares regression lines fit through the data, while the shaded blue region around them illustrate the 95% confidence interval for the regression estimate. The data underlying the plots in (C) and (D) can be found in https://datadryad.org/dataset/doi:10.5061/dryad.z8w9ghxmw.

frames with $S = 1, 1.5$, and 2. The corresponding 3D reconstructions of the comb built on these frames are presented in Fig 3B, which are the result of scanning sections of the combs highlighted with red boxes in Fig 3A. Tracing the plastic edges shown in blue and the comb

in dark yellow across the three samples displayed in Fig 3B reveals a consistent behavior: bees follow the printed foundations, regardless of increasing cell sizes.

The distribution of the size of the cells built on frames with foundations with $S = 1, 1.5, 2$ is shown in Fig 3C, with their detailed statistics shown in Table 1. A linear least-squares regression line that fits the cell size data, using normalized cell size as the response and the foundation size as the predictor, has a positive slope that is statistically significant ($\beta = 0.89, P < 10^{-5}$), demonstrating an increasing trend of built cell sizes with the foundation size. Moreover, Fig 3C suggests that this increasing trend in cell sizes is monotonic, with the strength of the trend highlighted by a high correlation coefficient (Pearson's correlation coefficient $\rho = 0.97, P < 10^{-5}$), indicating a strong positive relationship between the foundation and the final cell size. This provides strong evidence of the bees' ability to effectively utilize the printed foundations by adjusting honeycomb cell shapes and sizes in response to structural changes. According to our observations (which are consistent with previous findings [22,34–36]), these larger cells (built on foundations with $1 < S \leq 2$) are not suitable for raising workers, which are normally raised in cells built with $S \approx 1$. Therefore, these larger cells are either used for raising drones or storing honey. See S2 Fig for some examples of the usages of these cells in our experimental frames. For context, the average drone cell size [37] when normalized is equal to $S = 1.61$.

As shown in Fig 3D, we find that bees significantly increase the tilt of cells as the foundation sizes increase. Previous studies exploring the natural tilt of honeycomb cells have indicated that a standard worker cell tilts up to 13° [23,37,38]. It has also been established that the direction of this tilt is influenced by gravity: bees consistently tilt cells upward [26,39,40]. In our experiments, we note that cells on the control frames with $S = 1$ are tilted, on average, by 8.74° against the direction of gravity. For comparison, we computed the average tilt angle of natural drone cells to be 20° (see S3 Fig for more information). The tilt of the drone comb is measured manually using images of natural drone comb built on a flat surface, such as S3 Fig. The mean angle of tilts, their 95% confidence intervals and standard deviations for each foundation size $1 \leq S \leq 2$ are shown in Table 1.

Fig 3D shows the tilt distribution of the comb cells built on various foundation sizes, along with a best-fit linear regression line. It is evident that there is a gradual increase in cell tilts with increasing values of $S$, with a linear least-squares regression line fit to the angle data (response: angle of tilt, predictor: foundation size) showing a statistically significant positive slope ($\beta = 17.36, P < 10^{-5}$). Although the linear positive trend is strong (Pearson's correlation coefficient: $\rho = 0.84, P < 10^{-5}$), this relationship is not strictly monotonic. In particular, cells built on foundations with $S = 1.75$ are, on average, less tilted compared to those with $S = 1.5$ (under a one-tailed two-sample Welch's unequal variances t-test, $t = -8.04, P < 10^{-5}$). This trend suggests that cell size is not the only influential parameter and the tilt may be influenced by various structural factors of the hive, such as cell position on the frame, usage, or content.

**Table 1. Mean and standard deviation of cell sizes and tilts measured across various printed cell sizes (S)**

| Foundation cell size | Normalized cell size | | | Tilt [degrees] | | |
|---|---|---|---|---|---|---|
| | Mean ($\mu$) | Std. dev. ($\sigma$) | 95% CIs | Mean ($\mu$) | Std. dev. ($\sigma$) | 95% CIs |
| 1 | 1 | 0.03 | [1.00, 1.00] | 8.74 | 2.34 | [7.75, 9.73] |
| 1.25 | 1.19 | 0.09 | [1.19, 1.20] | 9.45 | 2.03 | [8.44, 10.46] |
| 1.5 | 1.49 | 0.06 | [1.48, 1.49] | 22.64 | 0.86 | [22.18, 23.10] |
| 1.75 | 1.6 | 0.1 | [1.60, 1.61] | 16.71 | 2.82 | [15.20, 18.21] |
| 2 | 1.89 | 0.07 | [1.89, 1.90] | 27.1 | 1.72 | [26.10, 28.09] |

Previous studies have suggested various hypotheses to explain the natural tilt of honeycomb cells. Mullenhoff [41] posited that this tilt prevents honey outflow. However, a recent study by Oeder et al. [38] challenges this idea, proposing instead that the tilt redistributes weight onto the midwall, potentially increasing the comb's carrying capacity. Additionally, in our experiments, we find that the amount of tilt effectively reduces the cross-sectional area of the cell, when considered perpendicular to the walls. When we cut the cells of the sample built on foundations with $S = 2$ using a plane perpendicular to the walls at the top of the sample, we find that the cross-sectional area of the hexagons in that layer is, on average, 0.77 times the area of the printed hexagons at its bottom. In other words, the cross-sectional area is equivalent to $S = 1.54$, a significant reduction compared to the size of the provided foundation $S = 2$. Moreover, these tilted cells also have a greater depth, defined along the direction of the walls, for a given total volume of the cell. In general, the angle of tilt could represent a compromise between maximizing volume, ensuring structural stability, and optimizing construction efficiency. See S2 movie–S6 movie for the 3D reconstruction of all the samples built with tilting cells. Additional details on the movies are provided in Table C in S1 text.

## Layering

Lastly, we explore cells built on frames with areas three times as large as than worker cells (*i.e.,* $S = 3$), which leads to a new mode of construction; see Fig 4. Unlike the previous case, bees do not adapt the provided pattern simply by constructing larger and more tilted cells. Instead, they use the vertices of 3D-printed cells as the foundation for a new layer of cells, forming a two-layer structure; see Fig 4A. This ability to build cells that are centered and supported by the edges of previously built cells reflects their natural behavior. Fig 4B shows a piece of a two-sided comb from a hive without foundation, where the front side is built upon the structural support of the pre-existing lattice on the back. The geometry of the 3D-printed frames in our experiments resulted in the emergence of an additional cell, located at the center of the printed hexagons. This cell extends to the base of the 3D-printed frame, merging with the printed hexagon. The positioning and structure of the cells are more clearly illustrated in the 3D view of the comb shown in Fig 4C, highlighting how the honeycomb cells (dark yellow) are built atop the plastic edges (blue).

Fig 4D depicts the distribution of normalized cell sizes of cells built on foundations with $S = 3$ and the reference cells ($S = 1$), with their means and standard deviations included. We find that the mean normalized cell size of cells built on foundations with $S = 3$ is 0.06 units smaller than the mean of the normalized reference cells, whereas these cells are 2.06 units smaller than the printed cells in the $S = 3$ frames, showing that cells built on foundations with $S = 3$ are much closer in size to reference worker cells than to their printed foundations. Typically, these cells are smaller than reference worker cells ($\mu_{S=3} = 0.94$, $\mu_{S=1} = 1$, $t = -26.67$, P < $10^{-5}$ for a one-tailed two-sample Welch's unequal variances t-test), while the variation in their normalized cell sizes is wider than that of the reference cell size distribution ($\sigma_{S=3} = 0.06$, $\sigma_{S=1} = 0.03$, $W = 285.59$, P < $10^{-5}$ for the Levene test to compare variances). The similarity in size of the cells built on $S = 3$ foundations to that of the reference cells suggests that the two-layer strategy employed by bees effectively transforms the printed foundation of $S = 3$, which is not suitable for any purpose in the hive, to $S = 1$, which is the most commonly used.

Overall, this symmetrical structure shown in Fig 4A, is composed of six newly built hexagons layered over the original foundation, forming another hexagon in its center. Consequently, as illustrated in the zoomed-in image in Fig 4A, each gray hexagon in the top layer derives approximately one-third of its area from the 3D-printed (blue) cell and the remaining two-thirds from the neighboring cells on which it is built. Another notable aspect of this

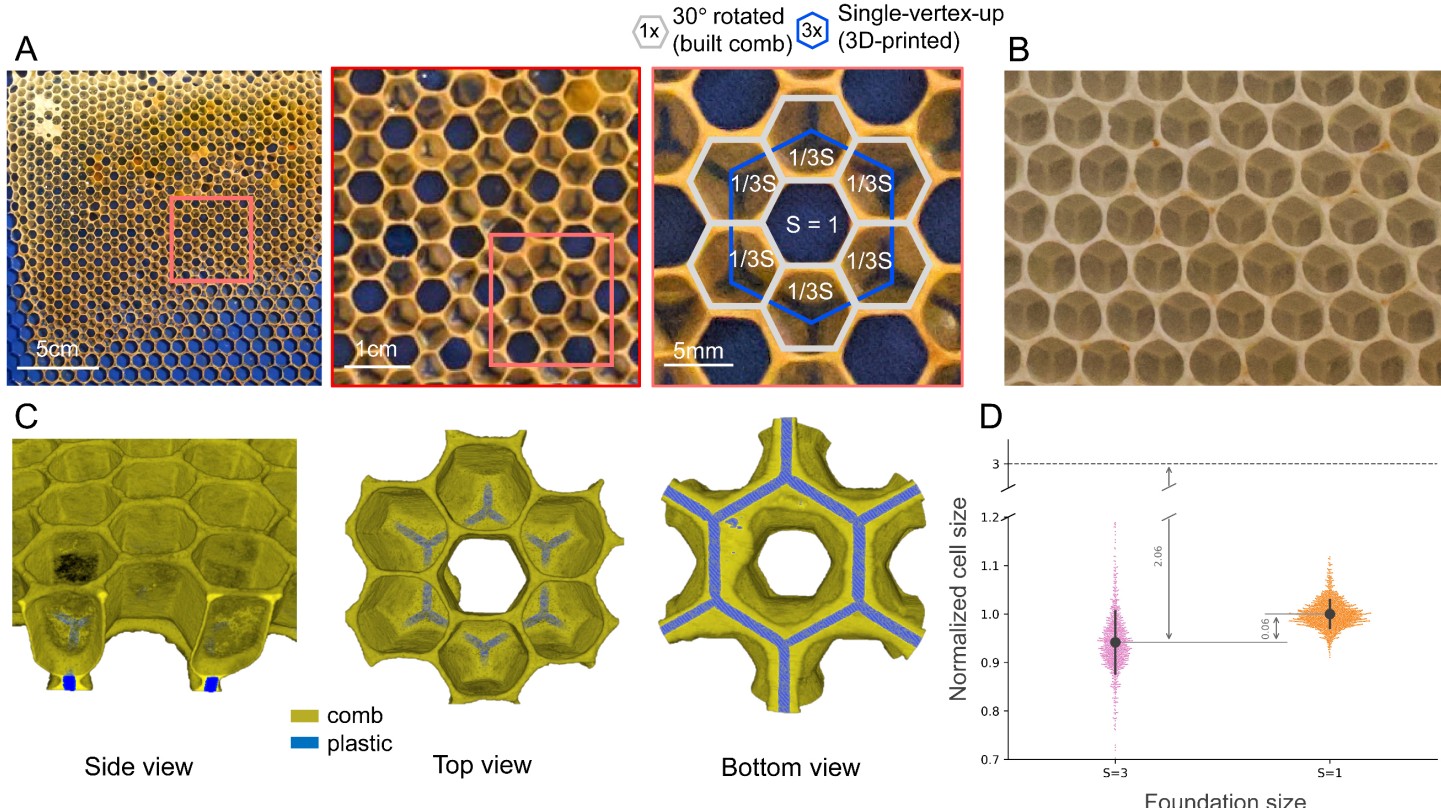

**Fig 4. Results of the comb built on samples with** $S = 3$ **foundation.** A) Image of the comb built on the frame with $S = 3$, followed by the zoomed in view of the area highlighted with a red square. B) A piece of natural two-sided honeycomb. C) Three views of the 3D reconstruction focusing on one printed hexagon and the surrounding cells using X-ray data, showing the comb in dark yellow and the plastic in blue. D) Comparison of the normalized cell size distributions of cells built on foundations with $S = 3$ in pink and cells built on the reference frames with $S = 1$ in orange. The gray circles within the distributions indicate the mean for the distributions and the gray errorbars depict the standard deviations. The deviation in the mean cell size of cells built on $S = 3$ foundations is shown relative to both the reference cell and the printed $S = 3$ cell. Note that a section of the vertical axis has been truncated for better visibility of the necessary details. The data underlying the plot in (D) can be found in https://datadryad.org/dataset/doi:10.5061/dryad.z8w9ghxmw.

structure is the change in lattice orientation during this geometric transformation. Although we printed the hexagons on the foundation with a single-vertex-up orientation, the honeycomb lattice built by the bees exhibits a 30° rotated structure (as shown by the blue and gray hexagons in Fig 4A). While the former orientation is commonly used in commercially available foundations and it is the orientation most observed in naturally built comb, the latter has been found to be approximately 30% stronger structurally [19]. Further increase in the printed cell sizes (*e.g.,* $S = 4$) did not produce any new patterns. We will discuss more details about all the tested patterns in the following section.

## Discussion

The flexibility of comb building behavior in bees has been previously discussed as a result of stigmergy [42] combined with a highly rich behavioral repertoire owned by bees [43]. Our work highlights how honeybees, like many other architecturally proficient animals, do not follow a fixed blueprint to build their nest. Instead, they employ a surprisingly diverse repertoire of construction modes and adjustments in response to varying environmental conditions such as changes in foundation cell sizes. This collective flexibility enables them to generate

adaptive solutions to local problems, contributing to the overall resilience and functionality of the colony's 'superorganismal' structure. The range of construction modes we documented in response to specific environmental conditions (*i.e.*, 3D-printed cell size) indicates a sophisticated level of problem-solving and plasticity that parallels adaptive building in ants, spiders, termites, weaver birds, etc. The observed variety of solutions might suggest cognitive planning [44,45], wherein bees potentially maintain a mental template of the desired outcome and select appropriate building modes based on foundation geometry. However, addressing the cognitive aspects of forward planning at the colony level would require targeted experiments that are beyond the scope of this paper. Here, our primary focus remains on how bees demonstrate multiple effective strategies for managing geometric challenges related to cell size. In particular, the merging, tilting, and layering adaptations we observed represent strategies to preserve structural integrity and functional cell geometry under constraints–similar to how spiders adjust web tension in response to wind [46,47], or how termites reinforce sections of their mound to effectively flush out $CO_2$ and ventilate the nest [48].

We examined how bees respond to foundation cell sizes within the range of $0.5 \leq S \leq 4$. Notably, when $S = 0.5$, or $S = 4$, bees do not effectively use the printed foundation patterns. For $S = 0.5$, they either disregard the printed pattern entirely or fill the small cells with wax (see Fig 1D), enabling them to initiate comb construction on a flat foundation. This behavior resembles the nesting practices of wild honeybees found in tree cavities with numerous small cracks and crevices [49], where they apply tree resins (propolis) to seal, coat, and create a smooth, waterproof base that protects against mold and bacteria [20]. At the oposite extreme, when $S = 4$, bees are similarly unable to use the printed foundations to build a uniform structure. Instead, their comb appears to be randomly constructed in all directions, sometimes growing outwards perpendicular to the frame (see Fig 1D). For intermediate sizes such as $S = 2.5$ and $S = 3.5$, we observed inconsistent mix of tilted and layered hexagonal cells (see S4 Fig for examples), suggesting more difficulty to generate uniform structures using a single mode of construction at these scales.

For the remaining foundations tested ($0.75 \leq S \leq 3$) our results show different mechanisms by which honeybees adapt regular hexagonal foundations of different sizes to respond to the imposed structural changes while satisfying colony needs. In the cases of $S = 0.75$ and $S = 3$, the 3D-printed cell sizes are quite different from those naturally used by bees, requiring notable adjustments to build a new lattice. In the case of smaller cells, the size mismatch between both layers is resolved by effectively blocking some 3D-printed cells, redistributing their space among adjacent ones. This can be understood as the discrete version of systems in which geometric mismatch between two layers is resolved through localization of deformation, from cracks in drying films or mud [50,51] to ridges in pre-stretched elastic bilayers [52,53], which result in similar patterns to those created by the blocked hexagonal cells. For $S = 3$, bees use the plastic edges of the printed frame as scaffolds to build a new layer of hexagonal lattice, centered on the edges and vertices of the foundation. This reconstructed lattice has a cell size close to $S = 1$, closely mirroring comb construction in African honeybees documented in [19]. While Hepburn described this middle cell as a "false" cell not easily noticed in the completed comb, our observations suggest that the broad base of this central cell can serve as an additional storage space for honey and pollen. In the range $1 \leq S \leq 2$, bees extend the provided lattice, constructing larger cells as $S$ increases. These larger cells are often more tilted, suggesting a structural or functional adaptation linked to increased volume. For a comprehensive comparison of the cell sizes built on all provided foundations, please refer to S5 Fig. These findings not only deepen our knowledge of honeybee ingenuity but also contribute to a broader understanding of how complex, functional structures can emerge from

adaptive behavioral responses. Our work envisions honeycombs not just as static structures, but as dynamically adapted solutions to a variety of environmental challenges.

In our previous work [25], we introduced a 2D model capable of replicating the patterns and defects in the comb structure arising from various geometric frustrations. Building upon the identification of novel building modes, particularly the two-layered pattern in our current study, we are motivated to extend our 2D model into a 3D framework. This extension will enable us to delve deeper into understanding the potential costs and benefits associated with such structural adaptations, which could be leveraged in the design of novel lightweight structures conforming to complex irregular geometries. Furthermore, given the bees' demonstrated capacity to construct and utilize large tilted cells, our future work involves rationalizing the relationship between cell size and angle, starting by examining the mechanical stability of the heavily tilted cells and the fluid mechanics that govern honey outflow within these structures.

Beyond structural considerations, future research should aim to examine the underlying behavioral algorithms and sensory cues driving these adaptive construction choices in honeybees, potentially drawing parallels with decision-making processes in other animal builders. Investigating the response of bees to foundations with lattices introducing some added disorder or defective patterns such as non-hexagonal cells could further reveal how they interpret and resolve them. Comparative studies across social insects and other animal builders could uncover convergent evolutionary pathways for adaptive construction behaviors, enhancing our understanding of the complex dynamics and functional significance of these remarkable living structures within a broader ecological and evolutionary context.

## Methods

To collect data about the structure of the comb, we establish a set of behavioral assays combined with X-ray imaging. The focus of this study is on the cell size ($0.5 \leq S \leq 4$) which is varied systematically using 3D-printed frames so that its effect can be isolated and quantified. The design work for preparing the 3D-printed frames is conducted using the SolidWorks 2019 CAD design software. All of our experiments are performed using colonies of European honeybees *Apis mellifera* L. at the Peleg lab apiary in Boulder, Colorado. We use langstroth hives with 19 in (480 mm) frames, with $9\frac{1}{8}$ in (230 mm) depth, and $1\frac{3}{8}$ in (35 mm) width. Each frame consists of a pair of 3D-printed plates of the size $212 \times 212$ mm that are placed opposite to each other. For our control set with $S = 1$, each hexagonal element on the 3D-printed plates has a side-length of 2.7 mm and outer area of 18.94 mm$^2$ (including the cell walls). All the other configurations are designed relative to the area of our control set. To confirm the observed modes of construction, we collected at least three repetitions of samples for each cell size, $S$. To account for at least three repetitions of each of the configurations under study, we scatter a total of 50 3D-printed experimental frames into ten honeybee colonies and fill the rest of the space in the hives with normal frames with full plastic foundations. In general, each colony has half 3D-printed frames and half normal frames. During the season, we often move the experimental frames between the colonies, so we do not control the construction behavior at the colony level.

To stimulate comb building, we apply a fine layer of natural beeswax to the printed foundation. The comb construction progress is monitored through regular photography sessions, capturing images of both sides of the experimental frames. To ensure minimal disruption during photography, we set up our photography station within the apiary, allowing us to quickly return the frames to their original locations inside the hives. Once the experimental

frames are fully built, they are carefully extracted from the hives, and kept in the lab for further analysis. We obtain high-quality images of the comb using a Nikon DSLR camera and controlled lighting, and a black background. The camera is mounted on a fixed metal structure 2 m from the comb to avoid distortion at the edges of the image and to keep a constant pixel-to-millimeter relationship. At this distance and camera resolution, 1 mm = 15 pixels.

To further analyze the honeycomb structures obtained after the completion of the experiments, we used 3D X-ray microscopy (XRM) to scan 5 cm × 5 cm sections of the comb constructed on our experimental frames. This provides a series of high resolution cross-sections of pieces of the comb that can be combined to generate a 3D virtual model. All tomography scans are performed using $\mu - CT$ ZEISS Xradia 520 Versa (Carl Zeiss X-ray Microscopy Inc, Pleasanton, CA, USA). The X-ray source is set to voltages in range 80-90 kV and power 7 W, while the detector camera is set with an exposure time of 2 seconds for each of the 2034 images captured by optical magnification 0.4. Image processing and segmentation on the X-ray data are conducted using Dragonfly (Object Research Systems, Montreal, QC, Canada). See the "X-ray Data Analysis" section in the Supporting information for a detailed account of our processing pipeline.

## Supporting information

**S1 Text. Additional information on X-ray data analysis.** Supplementary Figures, and Supplementary Tables. **Fig A.** Processing of X-ray tomography scans. **Fig B.** Calculation of cell tilt angle. **Table A.** Cell size statistics across various printed cell sizes. **Table B.** Covered cell statistics for $S = 0.75$. **Table C.** Details on supplementary movie files.
(PDF)

**S1 Fig. Honeycomb on commercial frame.** Honeycomb built on one of the commercial frames inside our hives using the merging strategy. A section of comb is magnified to show the occasional combination of the plastic edges on the foundation to build larger cells on top of them.
(TIF)

**S2 Fig. Samples with $S > 1$.** Sample frames with larger cell size foundations are either used for raising drone brood or honey storage.
(TIF)

**S3 Fig. Drone cells.** Honeycomb containing drone brood built on the flat side of the experimental frames. This is used to compute the value of 20° as the natural tilt of the drone comb, with a standard deviation of 2.43° and 95% CI: [18.86, 21.14].
(TIF)

**S4 Fig. Samples with $S = 2.5$ and $S = 3$.** Samples of acquired data on frames with $S = 2.5$, and $S = 3.5$ show a combination of tilted or layered modes of building on random sections of the frames.
(TIF)

**S5 Fig. Cell size distribution.** Comparison of cell size distribution of honeycomb built on all of the 3D-printed frames. The given cell sizes (S) are shown with different colors in the figure legend. The data underlying this plot can be found in https://datadryad.org/dataset/doi:10.5061/dryad.z8w9ghxmw.
(TIF)

**S1 Movie. 3D scan of sample with $S = 0.75$.**
(MP4)

**S2 Movie. 3D scan of sample with $S = 1$.**
(MP4)

**S3 Movie. 3D scan of sample with $S = 1.25$.**
(MP4)

**S4 Movie. 3D scan of sample with $S = 1.5$.**
(MP4)

**S5 Movie. 3D scan of sample with $S = 1.75$.**
(MP4)

**S6 Movie. 3D scan of sample with $S = 2$.**
(MP4)

**S7 Movie. 3D scan of sample with $S = 3$.**
(MP4)

## Acknowledgments

XRM data collection and parts of the analyses were performed at MIMIC facility, at CU Boulder (RRID:SCR_019307). We thank Dan Larremore for suggestions and help with data processing, Seneca Kristjonsdottir and Christopher Borke for bee management, Adrian Gestos for XRM training and trouble shooting, Paul Bontempo, Richard Terille, Anna Rahn, and Anna Simone for their assistance with data collection and organization.

## Author contributions

**Conceptualization:** Golnar Gharooni-Fard, Orit Peleg, Francisco López Jiménez.

**Data curation:** Golnar Gharooni-Fard, Chethan Kavaraganahalli Prasanna.

**Formal analysis:** Chethan Kavaraganahalli Prasanna.

**Funding acquisition:** Orit Peleg, Francisco López Jiménez.

**Investigation:** Golnar Gharooni-Fard.

**Methodology:** Golnar Gharooni-Fard, Orit Peleg.

**Project administration:** Orit Peleg, Francisco López Jiménez.

**Software:** Golnar Gharooni-Fard, Chethan Kavaraganahalli Prasanna.

**Supervision:** Orit Peleg, Francisco López Jiménez.

**Writing – original draft:** Golnar Gharooni-Fard, Chethan Kavaraganahalli Prasanna.

**Writing – review & editing:** Orit Peleg, Francisco López Jiménez.

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
