## [Editor Report · Decision Letter 0]

17 Jun 2024

Dear Dr Lopez Jimenez,

Thank you for submitting your manuscript entitled "Adaptive Cell Size, Merging, Tilting, and Layering in Honeybee Comb Construction" for consideration as a Research Article by PLOS Biology.

Your manuscript has now been evaluated by the PLOS Biology editorial staff, as well as by an academic editor with relevant expertise, and I'm writing to let you know that we would like to send your submission out for external peer review.

IMPORTANT: We think that your paper would be better considered as a Short Report. No re-formatting is required, but please can you change your article type to "Short Reports" when you upload your additional metadata (see next paragraph)?

Once your full submission is complete, your paper will undergo a series of checks in preparation for peer review. After your manuscript has passed the checks it will be sent out for review. To provide the metadata for your submission, please Login to Editorial Manager (https://www.editorialmanager.com/pbiology) within two working days, i.e. by Jun 19 2024 11:59PM.

Kind regards,

Roland

Roland Roberts, PhD

Senior Editor

PLOS Biology

rroberts@plos.org

---

## [Decision Letter · Decision Letter 1]

13 Sep 2024

Dear Dr Lopez Jimenez,

Thank you for your patience while your manuscript "Adaptive Cell Size, Merging, Tilting, and Layering in Honeybee Comb Construction" was peer-reviewed at PLOS Biology. Please allow us again to apologize for the long delay in sending the decision. It has now been evaluated by the PLOS Biology editors, an Academic Editor with relevant expertise, and by several independent reviewers.

In light of the reviews, which you will find at the end of this email, we would like to invite you to revise the work to thoroughly address the reviewers' reports.

As you will see below, the reviewers agree that the topic of your study is very interesting and that it is overall well done. However, they also raise a few concerns that need to be addressed, for example about the accessibility of the manuscript for a broader audience and missing statistical analyses.

Given the extent of revision needed, we cannot make a decision about publication until we have seen the revised manuscript and your response to the reviewers' comments. Your revised manuscript is likely to be sent for further evaluation by all or a subset of the reviewers.

**IMPORTANT - SUBMITTING YOUR REVISION**

*Re-submission Checklist*

*Published Peer Review*

*PLOS Data Policy*

*Blot and Gel Data Policy*

Sincerely,

Christian Schnell (on behalf of Roland, who is out of office this week)

Senior Editor

PLOS Biology

cschnell@plos.org

Roland Roberts, PhD

Senior Editor

PLOS Biology

rroberts@plos.org

REVIEWS:

Reviewer #1: Overall, I felt this study was very original, and offers a nice complement to the authors earlier PNAS article (Gharooni-Fard et al. PNAS 2022). The study presents novel findings on the adaptability of honeybees in constructing combs with varying cell sizes, and does so using a set of innovative techniques, such as 3D printing and X-ray Microscopy (XRM). This research is highly relevant and significant both for biologists interested in self organisation and collective behaviour as well as for engineers interested in bio-inspired designs. The manuscript is well-organized and clearly written, and makes complex experimental setups and results accessible to non-specialists. The methodological section is also sufficiently detailed to allow results to be reproduced by others. Overall, I think the results are of sufficient general interest to be able to recommend publication in PloS Biology.

General comments:

Abstract, introduction & discussion: I think the abstract, introduction and discussion would benefit from being pitched at a more generic level, e.g. also emphasizing the broader implications of the study in the field of collective behaviour and self organisation. For example, a discussion similar to that of Gallo & Chittka (PNAS, 2021, "Stigmergy versus behavioral flexibility and planning in honeybee comb construction") could be nice, along with a more broadly pitched introduction and discussion. My main concern is that the article should be placed better within the larger body of prior work on collective behaviour and self organisation. Even for the specific topic of this article, I think some extra articles could be worth citing : aside from the Gallo & Chittka PNAS commentary, perhaps also Gallo et al. J. Comp. Phys. A (2023), Krishna et al. Sci. Rep. (2022), Smith et al. PloS Biol. (2022) and Yang et al. PloS One (2022).

Methods & results: While the methodology is rigorous, the results are mainly described in purely descriptive terms and do not entail any detailed statistical methods. Some of the results on the presented frequencies of the different types of cells and patterns could benefit from a formal statistical analysis though, e.g. using linear mixed models (LMMs) or generalized linear mixed models (GLMMs) with the use of appropriate error structures (e.g. gaussian, binomial or multinomial) and the inclusion of random intercepts to account for colony-specific variation and/or overdispersion (through the inclusion of an observation-level random effect) (both colony-level effects and any overdispersion as well as the specific distribution of the data are currently all ignored). Additionally, including confidence intervals and effect sizes in the analysis would strengthen the statistical robustness of the findings. These analyses could be done e.g. using the lme4 and emmeans packages (for binomial GLMMs and Gaussian LMMs) or using mclogit::mblogit for multinomial GLMMs.

Minor comments:

* Lines 13-14: The sentence "For smaller cell size, bees occasionally merge cells to compensate for the reduced space" should be rephrased for clarity. Suggestion: "For smaller cell sizes, bees occasionally merge adjacent cells to optimize space utilization."

* Lines 45-47: Provide more details or examples of how the degree of adaptability and variability in honeycomb structure was previously assessed in 2D studies. Mention specific studies and their findings.

* Line 64: Clarify the specific contributions of Franklin et al. (2022) to set the context for the current study. For example, "Franklin et al. (2022) used XRM technology to describe the step-by-step process of honeycomb construction in natural settings, highlighting the intricate structure of honeycomb cells."

* Lines 170-172: Provide some extra literature citations supporting the fact that larger cells tend to be used for raising the larger drones or storing honey.

* Lines 130-136: Please use a proper statistical model (e.g. a binomial GLMM with colony included as a random intercept fir using glmer::lme4) to give 95% confidence intervals on that 25% proportion. You could also use the emmeans package to test that the observed ratio does not differ significantly from the expected 25%.

* Lines 143-144: Again, use a proper statistical model to describe these results (e.g. a gls generalised least squares model with an error structure that allows for differences in variability across both groups, and use emmeans to report effect sizes and 95% confidence intervals). You could also compare the AIC of a gls model with identical or heterogeneous variances to show that a model with different variances fits better.

* Lines 174-175: Again use proper statistical methods here as opposed to just describing the results.

* Lines 183-184, lines 187-189: Fit proper statistical models to these data (e.g. a linear mixed model or generalised linear mixed model) and report effect sizes and 95% confidence intervals using emmeans. Nonlinearity could be included via a natural cubic spline (ns(), e.g. with df=2) term in the model.

* Lines 222-224: Fit proper statistical models (e.g. a linear mixed model or generalised linear mixed model) to these data and report effect sizes and 95% confidence intervals using emmeans.

Reviewer #2 (Amir Haluts): The authors report interesting results regarding the way bees reconcile, in their adaptive hive building, suboptimal foundational geometry imposed upon them. I found no issues with the experiments, methodologies, and analyses conducted to produce the results. This study appears to be of interest to a rather wide audience and suitable for the scope of PLOS Biology. The paper is fairly well written, although I think the authors could make the writing slightly more consistent (in terms of terminology used) and cohesive --- which would assist the reader's understanding of the main findings in this short report. Specifically, I would like the authors to address the following minor comments:

Lines 37-38:

Instead of "...differing combination of cell sizes, alignments, etc. at the boundaries", I suggest <differing geometries of the cells at the boundaries (notably in terms of size and alignment)>.

Line 74:

"...derived from the value of the diagonal..." --- <length> seems to be a better word here.

Lines 74-75:

"...perfect hexagon..." --- <regular> hexagon? (keep it consistent throughout the paper).

Line 96:

"...overall structure of the comb in each category." --- It appears you are using both "modes" and "categories" to refer to the same thing in the same sentence. Then in the next sentence you use categories. Better keep it consistent and call it, say, <construction modes> from the beginning.

Lines 102-103:

"...faced with smaller cell sizes..." --- <foundational> cell sizes? There should be a clear distinction throughout the text between the 3D-printed cells provided as foundation and the wax cells built by the bees. Using just "cell" for both of them is confusing.

Lines 123:

"...axis" - <axes>

Lines 125-126:

"...adjacent plastic edges beneath the cells constructed on top of them." --- This sentence is somewhat convoluted. I recommend rephrasing it to increase clarity.

Line 130:

"The distribution of covered cells..." --- Again here, it will be helpful for the reader to consistently distinguish in the text between "foundational"\"printed"\"scaffold" cells and the cells built by the bees.

Lines 142-146:

While you provided a reasonable explanation for the higher standard deviation, an explanation for the clearly higher average (S = 1.12 vs. S = 1.04) is missing. Could this be explained?

Line 150:

Does "worker-size foundations" mean S = 1? Better make this clear.

Lines 163-165:

"...Fig. 3B reveals a consistent behavior: bees construct honeycomb cells along the edges of plastic foundations, regardless of the increasing cell sizes observed in the samples." --- The purpose of this statement, and its relation to the next statement, is unclear.

Line 168:

"...a clear positive correlation between the foundation and the final cell size." --- What's the correlation coefficient?

Lines 170-172:

"According to our observations, these larger cells that are too large for raising workers, are either used for raising drones or storing honey (see Fig. S4 for some examples)." --- From the distributions in Fig. 3C, it seems that S = 1 lies outside the distribution of the cells built by the bees for all the foundations with S > 1. Namely, it appears that all cells in these cases are "too large" for raising workers. Are workers not raised in these hives at all? Could you make this statement clearer in these regards?

Lines 174-175:

"While the dashed line aligns well with the S = 1.75 and closely follows the S = 1.5 distributions, it is quite far from S = 2." --- I think it is more accurate to say <While the dashed line matches the mean of the S = 1.75 distribution ({provide the mean}) and overlaps with the S = 1.5 distribution, it does not overlap with the S = 2 distribution.>

Line 187:

"...this relationship is not strictly linear." --- <monotonic> seems to be more accurate than linear here.

Lines 187-188:

"Notably, cells built on foundations with S = 1.75 are, on average, smaller than those with S = 1.5..." --- The numbers in Table 1 indicate the opposite - the cells built on S = 1.75 are larger than those built on S = 1.5 (S = 1.67 vs. S = 1.55 on average, respectively). Did you mean that they have <smaller average tilts>?

Line 190:

"...a non-linear relationship..." --- I'm not sure "non-linear" is the best term here. Non-monotonic seems to be a better term.

Caption of Figure 3:

"A) The 2D images of the frame with S = 1, 1.5, and 2. B) The 3D reconstruction of the sections of the comb, highlighted with a red box in panel A, using X-ray data, segmented to show comb in dark yellow and the plastic in blue." --- Which is which? Indicate on the panels S = 1, S = 1.5, and S = 2.

Caption of Figure 3:

"...frames with larger foundations" --- <frames with 1 <= S <= 2>.

Lines 199-205:

So, are you implying that the bees are using tilting as another compensation mechanism for suboptimal foundations (this time for S > 1)? If so, I suggest making this statement explicitly.

Line 212:

"...they use the corners..." --- I suggest using <vertices> instead of corners (this is a better term, and also keeps it consistent with the discussion).

Line 214:

Again, use <vertices> instead of corners.

Line 223:

"...but quite similar to the cell sizes..." --- I suggest <but has a similar mean cell size>.

Lines 232-235:

"It is worth noting that, while we printed the three-times-larger hexagon with a vertical orientation, with its side walls perpendicular to the top bar, the resulting honeycomb lattice built by bees features a horizontal orientation, with side walls parallel to the top bar." --- I found it difficult to understand this sentence. What is the "top bar"? What do you mean by "vertical" and "perpendicular" orientations? Please clarify.

Line 242:

"...when S = 0.5, or S = 4" --- I saw no reference in the results to these sizes. Maybe worth mentioning briefly the outcome of S = 0.5 before elaborating on S = 0.75, and the outcome of S = 4 after elaborating on S = 3.

Lines 282-294:

In journal paper discussions, I prefer statements like this phrased as "what can be done (by anyone)" and not as "what we will do", but this is somewhat a matter of taste.

Reviewer #3: The authors of this paper have used some recent technology to identify how bees respond to variation in surfaces when building their combs. While I think that there is some really interesting and valuable data presented here, the focus of the paper is, in my view, rather too focussed on the technology than on the biological import of the data. In sum, the narrative here is dense and rather too distant from the biology to see how the authors have added substantially to what we know already. What would be useful (at least for this reader) would be for the authors to make their story more biologically accessible, otherwise there seem likely to be few who will see what they have added. It would help to be clearer about the problem and what is already known, and what questions this work specifically addressed. The authors currently describe the background as not having yet provided a thorough investigation into the extent of the adaptability of bees constructions and that they "shed light honeybees' adaptive comb construction strategies with potential to find applications in additive manufacturing, bio-inspired materials, and entomology." I am not really sure what this means: it currently seems too vague to be an especially valuable set of outcomes.

The data seem sound enough but it is the biological interpretation that needs work and I think the authors need to make that clearer and more compelling before the substance of their work will be clear.

Reviewer #4: The paper studies how honeybees 'adapt' their honeycomb building strategy to 'boundary conditions' in the environment. The authors use a clever 3D printing set-up as a 'seeding' of the honey comb pattern, and let the bees build upon it. Figure 1 summarizes the main result : for small and big pattern, the bees essentially neglect the prepattern, but for intermediate seeds, three behaviours are observed. For slightly smaller pattern than the 'natural' one, a 'merged' pattern is observed, where the prepattern is used as a template for merging cells to reach the normal size.

For prepatterns slightly bigger (up to 2x) than the normal pattern, a 'tilted' pattern is used where, remarkably, the size distribution of the final cell sizes essentially follows the prepattern, but cells are constructed in a tilted way.

When the prepattern is 3x as big as the 'normal' size, a clever 'layered' strategy is used, where, by and large, the printed pattern is used as a template for cells of normal size, centered at each of the 'corner' of the prepattern.

The study is very interesting and well-written, and I think it would be of interest for a general biology audience.

I have the following comments/questions :

- For the merging case (Fig 2), is it possible to take the 3D x-ray, define cell size at a distance d from the prepattern, and to plot, say, the average cell size as a function of d ? (basically along the coordinate X shown in 2F,H). I am wondering if the transition from 0.75 to 1 is continuous, or if there is a kind of distance threshold from which all cells get to size 1.

- For the same experiment, it is also interesting that most cells have in fact much bigger normalized size than 1. It is not clear to me why we see this : it suggests that the cells are first increased in size before being divided. Can the authors comment more on this ?

- It seems to me that there are some 'magic numbers' (due to topology), where some special strategies might be used, and I am wondering if the authors could comment on this. For instance, it is striking to me that for a 1.5 size, the distribution of cell sizes is much narrower, and the tilt is much bigger than 1.25 and 1.75. In fact it seems closer to the natural drone comb.

- Along the same line, for the layered strategy, 3 seems to be such a 'magic number', since the bees can exactly fit normal size hexagons at each corner. So by comparison, I think it would be good to discuss and quantify a bit more in the main text the cases where the geometry can not be easily accommodated. The 2.5 case should be especially interested, the authors mention there is a mix of titled and layeded hexagonal cells, but how 'mixed' is it ? There could also be an interesting transition here to study, maybe for 2.8 the pattern is completely disordrered, and at 3 one gets something very regular

- Along the same lines, I am wondering what happens if some 'disorder' is introduced in the prepattern. For instance, one could imagine more irregular hexagons, or with topological defects where from time to time 5-sided or 7-sided cells are introduced. The building strategy might changes in response to this (maybe defects are kept, or maybe transiently bigger cells are build to implement merging later). I have no good sense of how difficult it would be to do those experiments, but they could also be very informative.

---

## [Decision Letter · Decision Letter 2]

9 May 2025

Dear Dr Lopez Jimenez,

Thank you for your patience while we considered your revised manuscript "Adaptive Cell Size, Merging, Tilting, and Layering in Honeybee Comb Construction" for publication as a Short Reports at PLOS Biology. This revised version of your manuscript has been evaluated by the PLOS Biology editors, the Academic Editor and the original reviewers.

Based on the reviews, we are likely to accept this manuscript for publication, provided you satisfactorily address the remaining points raised by the reviewers and the following data and other policy-related requests.

IMPORTANT - please attend to the following:

a) Please could you change your title to make it more explicit, declarative and appealing? We suggest "Honeybees adapt to inappropriate comb cell size by merging, tilting and layering their construction"

b) Please attend to the remaining requests from reviewers #1 and #3. I asked the Academic Editor about reviewer #3's comments, and s/he said "I guess what reviewer #3 has in mind is that the manuscript is a bit 'inward-looking' in that it's pitched inside the literature on honeybees, and it would be good to zoom out a bit and embed it a bit more widely in the literature on animal architectures (e.g. bird nests, etc). So yes I think it would be good to ask the authors to fix this in the revised version."

c) Please address my Data Policy requests below; specifically, we need you to supply the numerical values underlying Figs 2K, 3CD, 4D, S1BC, S7, either as a supplementary data file or as part of your Dryad deposition (assuming it's not already in the Dryad depo - that looks fairly raw to me).

d) Please cite the location of the data clearly in all relevant main and supplementary Figure legends, e.g. “The data underlying this Figure can be found in S1 Data” or “The data underlying this Figure can be found in https://datadryad.org/XXXXXX
https://zenodo.org/records/XXXXXXXX"

e) Many thanks for roviding the code in Github; however, because Github depositions can be readily changed or deleted, please make a permanent DOI’d copy (e.g. in Zenodo) and provide this URL.

We expect to receive your revised manuscript within two weeks.

*Published Peer Review History*

*Press*

Sincerely,

Roli Roberts

Roland Roberts, PhD

Senior Editor

rroberts@plos.org

PLOS Biology

DATA POLICY:

[Figs….]

CODE POLICY

DATA NOT SHOWN?

REVIEWERS' COMMENTS:

Reviewer #1:

General assessment

The authors have substantially improved the manuscript. The Introduction now situates the work firmly within the literature on stigmergic self organisation and collective construction, and the Discussion highlights the relevance for both evolutionary biology and bio inspired engineering. Most textual and citation issues raised previously have been resolved. Importantly, the authors augmented the Results section with quantitative analyses and confidence intervals.

Nevertheless, the statistical treatment still falls short of current best practice. Cell level observations arising from multiple frames and colonies are analysed with single level t and z tests that ignore non independence. Analysis of these data via mixed effects models would be recommended to obtain valid standard errors and to quantify colony to colony variability. Effect sizes should then be reported via estimated marginal means or contrasts with 95 % CIs.

With the mixed effects re analysis in place—together with minor editorial polishing—the manuscript will in my view meet the standards of PLOS Biology. I therefore recommend minor revision.

Specific comments

1. Statistics

- For every response variable analysed at cell level (covered cell proportion, cell area, tilt angle, etc.), it would be recommended to re fit a GLMM/LMM that includes colony (or colony × frame) as a random intercept and, where relevant, an observation level random effect to account for over dispersion. Subsequently, fixed effect estimates & 95 % CIs should be reported, alongside likelihood ratio or Wald p values. The packages that I recommended earlier ( lme4, mclogit, emmeans) will handle this.

2. Clarify replication structure

Specify the number of biological replicates (colonies) per treatment and how technical replicates (frames) are nested within colonies.

3. Figures & tables

- Update Figs 2-4 to display model based estimates (mean ± 95 % CI) in addition to raw data.

- Ensure Table 1 lists mixed model estimates, not simple means.

4. Terminology & editing

- Define XRM on first appearance.

- Maintain "printed" vs "built" cell terminology consistently.

- Verify that all supplementary figures cited in the text are present.

5. Data & code

Data and code availability statements are appreciated; please ensure that the GitHub repository also reproduces the statistical (ideally mixed model) analyses.

Subject to these relatively minor yet important revisions, I look forward to recommending acceptance.

Reviewer #2:

[identifies himself as Amir Haluts]

I have carefully reviewed the authors' responses to my comments and the corresponding revisions made to the manuscript. I am pleased to see that the authors have addressed all of my suggestions thoroughly and thoughtfully.

The comprehensive revisions made by the authors in response to my and the other reviewers' comments have substantially improved the quality of the manuscript, which now presents a more cohesive, accurate, and convincing account of this interesting research on honeybee adaptive hive construction strategies.

I am therefore satisfied that this manuscript is now suitable for publication in PLOS Biology.

Reviewer #3:

I have just a brief view on this paper. I appreciate the work the authors have put in to improve the accessibility of their paper and I think they have done a pretty good job with that. However, in so doing, they have shown that they are not able to describe the importance of their work: "Our experimental results extend our understanding of honeybees' remarkable comb-building adaptability, providing detailed evidence of their diverse construction techniques when faced with varying foundation cell sizes. Our findings align with and strengthen previous research by introducing more variety of construction modes in response to specific environmental conditions."

And the last line is still: "These future directions aim to enhance our comprehension of the intricate dynamics underlying comb construction and functionality of these adaptive structures."

Reviewer #4:

I understand most of my comments can not be addressed without extensive new experiments. This is completely justifiable. As said in my previous review I think the paper is interesting and deserves to be published.

---

## [Editor Report · Decision Letter 3]

9 Jun 2025

Dear Francisco,

Thank you for the submission of your revised Short Report "Honeybees adapt to a range of comb cell size by merging, tilting and layering their construction" for publication in PLOS Biology. On behalf of my colleagues and the Academic Editor, Lars Chittka, I'm pleased to say that we can in principle accept your manuscript for publication, provided you address any remaining formatting and reporting issues. These will be detailed in an email you should receive within 2-3 business days from our colleagues in the journal operations team; no action is required from you until then. Please note that we will not be able to formally accept your manuscript and schedule it for publication until you have completed any requested changes.

IMPORTANT: I've asked my colleagues to include the following request alongside their own: Thank you for providing the raw data. However, we also need the numerical values directly underlying Figs 2K, 3CD, 4D, S1BC, S7, either as a supplementary data file or as part of your Dryad deposition. These are probably best provided as simple Excel files, with one clearly labelled tab per Figure panel. Also please cite the location of the data clearly in all relevant main and supplementary Figure legends, e.g. “The data underlying this Figure can be found in S1 Data” or “The data underlying this Figure can be found in https://datadryad.org/XXXXXX"

Sincerely, 

Roli

Senior Editor

PLOS Biology

rroberts@plos.org